# Advancing Personalized Medicine in Common Forms of Parkinson’s Disease through Genetics: Current Therapeutics and the Future of Individualized Management

**DOI:** 10.3390/jpm11030169

**Published:** 2021-03-01

**Authors:** Xylena Reed, Artur Schumacher-Schuh, Jing Hu, Sara Bandres-Ciga

**Affiliations:** 1Laboratory of Neurogenetics, National Institute on Aging, National Institutes of Health, 35 Convent Drive, Room 1A-211, Bethesda, MD 20892-3707, USA; xylena.reed@nih.gov; 2Serviço de Neurologia, Hospital de Clínicas de Porto Alegre, Rua Ramiro Barcelos 2350, Porto Alegre, RS 90035-003, Brazil; schuh.afs@gmail.com; 3Departamento de Farmacologia, Universidade Federal do Rio Grande do Sul, Rua Sarmento Leite, 500 sala 305, Porto Alegre, RS 90050-170, Brazil; 4Simpson Querrey Center for Neurogenetics, Ken and Ruth Davee Department of Neurology, Northwestern University Feinberg School of Medicine, Chicago, IL 60611, USA; jing.hu@northwestern.edu

**Keywords:** precision medicine, Parkinson’s disease, genetics, clinical trials

## Abstract

Parkinson’s disease (PD) is a condition with heterogeneous clinical manifestations that vary in age at onset, rate of progression, disease course, severity, motor and non-motor symptoms, and a variable response to antiparkinsonian drugs. It is considered that there are multiple PD etiological subtypes, some of which could be predicted by genetics. The characterization and prediction of these distinct molecular entities provides a growing opportunity to use individualized management and personalized therapies. Dissecting the genetic architecture of PD is a critical step in identifying therapeutic targets, and genetics represents a step forward to sub-categorize and predict PD risk and progression. A better understanding and separation of genetic subtypes has immediate implications in clinical trial design by unraveling the different flavors of clinical presentation and development. Personalized medicine is a nascent area of research and represents a paramount challenge in the treatment and cure of PD. This manuscript summarizes the current state of precision medicine in the PD field and discusses how genetics has become the engine to gain insights into disease during our constant effort to develop potential etiological based interventions.

## 1. Introduction

Personalized medicine, also referred to as precision or stratified medicine, is a medical model that uses an individual’s biological profile to guide decisions made in regard to the prevention, diagnosis, and treatment of a disease [1]. Based on each patient’s unique molecular makeup, clinical information and personal preferences, it aims to overcome the limitations of traditional medicine by providing better diagnoses with earlier intervention. Combining all of this individual data allows for more efficient drug development and the advancement of more targeted therapies, by selecting the optimal treatment for a specific patient. The genomic, epigenomic, transcriptomic, and proteomic profile of an individual plays a crucial role in understanding how well a patient will respond to a certain treatment.

In the Parkinson’s disease (PD) field, precision medicine is a nascent and exciting area of research that ultimately aims to achieve an appropriate disease-modifying treatment, with the right dose, at the right time in a specific patient. The link of PD to α-synuclein was the first decisive proof of a genetic defect leading to disease [2]. Later on, the first PD genome-wide association studies (GWAS) identified *SNCA* [3] as one of the major genes driving risk for sporadic PD, linking both familial and sporadic forms. Abnormal α-synuclein is a histopathological hallmark of PD patients, but also patients with other neurodegenerative conditions, collectively termed synucleinopathies, making this target promising. However, the fact that PD patients harbouring genetic defects in genes such as *PRKN* do not present with Lewy body pathology, strengthens the notion that distinct entities and multiple overlapping etiologies are at play.

So far, our limited understanding of how common forms of PD start and progress at the cellular and molecular level alongside the challenge of establishing methods for early preclinical diagnosis have hampered the development of PD modifying therapies able to prevent, stop or slow down the neurodegenerative process. However, the future holds promise. Using genetics to stratify patients can help predict success in the clinic, and drugs targeting proteins with a genetic connection to disease are more likely to be approved [4].

Clinical trials targeting genetic forms of PD, such as patients with variants in *LRRK2* and *GBA* have already been initialized, highlighting the rapid progress made in the field in the past two decades [5]. As we piece together the complex molecular puzzle of PD by unraveling the underlying pathophysiology, our hope is that novel etiological based therapies will emerge. More studies will need to be done to understand whether these therapies would be useful only for specific variant carriers or if they could also be beneficial in some forms of idiopathic PD.

On another front, drugs currently used that have significant side effects in some individuals could be used more wisely to obtain more benefits with fewer adverse events when guided by genomic information. However, identifying the right treatment for a specific patient remains a daunting challenge. PD is a widely heterogeneous disease, and numerous etiological subtypes might exist. Therefore, treating PD as one disease with a single solution will only lead to failure. Increasing evidence suggests that defining subclasses of PD and developing tools to predict the course of the disease has the potential to significantly improve cohort selection in clinical trials, reduce their cost, and increase the ability of such trials to detect treatment effects [6]. On the whole, pure monogenic forms of PD are rare and although variants in genes like *SNCA*, *PINK1*, *PRKN*, and DJ1 are well established causes of disease it would be difficult to collect enough patients to create an appropriately powered clinical trial in these populations. For this reason, in this review we will focus on more common forms of disease including those with variants in known risk factors, like *LRRK2* and *GBA*, as well as idiopathic forms of PD where the exact cause is not known but it is thought to be a combination of genetic and environmental factors. Current estimates of PD heritability have revealed that the contribution of genetic factors to PD phenotype is about 22% indicating that stratifying patients by genomic factors is possible [7,8].

Just as important as knowing the right drug is knowing the right time in disease development to provide treatment before irreversible brain damage occurs. With current diagnostic tools, by the time there is a clinical manifestation of PD, a substantial number of dopaminergic neurons have already been permanently lost, so even if the right therapeutic is applied to the right patient, it is too late for a full recovery of motor symptoms. Using personalized medicine to examine the specific genetic context can also help identify individuals at higher risk of developing PD before symptoms appear.

This manuscript summarizes the current state of the role of genetics in precision medicine in common forms of PD. We will discuss how genetics has become the engine to gain insights into PD etiology during our constant effort to develop potential etiological based interventions.

## 2. Genetics as a Tool to Improve Current Symptomatic Treatment

The symptomatic treatment available for PD targets the motor symptoms induced by the dopaminergic deficit due to the degeneration of the substantia nigra. Nevertheless, the disease affects other systems and regions in the brain, which leads to a myriad of levodopa-resistant motor and non-motor symptoms for which we do not have well-established pharmacological interventions. Despite this limitation, PD is the only neurodegenerative disorder with a symptomatic treatment that provides a substantial benefit. Since the introduction of levodopa in the 1960s, it has changed the natural history of PD and remains the gold standard of treatment [9]. However, the pharmacological response is variable and, as the disease progresses, higher doses of levodopa are required. Moreover, complications induced by chronic treatment can develop over time, including motor fluctuations and dyskinesia, which affect almost half of the patients after five years of treatment and nearly all in the long term [10,11,12]. This situation impairs the patient’s quality of life and demands more costly and complex therapeutic regimens.

Pharmacogenetics assumes that the variability in the pharmacological response observed in the clinic, can be partially explained by genetics, envisioning a scenario where a patient’s genotype can assist in drug prescription. It is speculated that genetics accounts for 60–90% of the variability in the pharmacokinetics and pharmacodynamics of antiparkinsonian drugs [13]. Despite this, there is a lack of studies with robust designs that enable strong pharmacogenetic recommendations for these drugs. Most of the pharmacogenetics studies in PD were conducted in a “pre-genomic” era when variants in candidate genes were nominated with a hypothesis-driven approach [14,15].

Polymorphisms in genes related to dopamine metabolism, like *COMT*, *MAOB*, *SLC6A3*, and *DRD2*, were the natural candidates. Several phenotypes related to drug effect were studied, including levodopa response, dyskinesia, sleep disturbances, and hallucination. For example, COMT V158M, a polymorphism that alters enzyme activity, was associated with levodopa and COMT inhibitor response, while variants in the *DRD2* gene were associated with levodopa-induced dyskinesia [16]. However, these studies had small sample sizes, lacked independent replication and did not correct for multiple comparisons. The variant selection was not consistent, and the outcome assessment varied among them preventing any clear pharmacogenetic recommendations for clinicians [14].

The next frontier is pharmacogenomics, which is based largely on the data provided by genome-wide association studies (GWAS). GWAS uses genotyping arrays to identify variants that are associated with a particular phenotype by comparing the frequency of thousands of variants between cases and controls. This approach can assess the effect of genetics on pharmacological variability for a particular trait (in this case, a specific pharmacological response) using a hypothesis-free strategy. Pioneers of this approach in the pharmacogenomics of PD were two studies conducted in the same cohort that evaluated the effects of caffeine and smoking in 1458 patients with PD and 931 healthy controls [17,18]. The authors reported a gene-caffeine and gene-smoking interaction on PD risk at the risk loci *GRIN2A* and *SV2C*, respectively. In another study, Ryu et al. performed a GWAS to evaluate motor fluctuation and levodopa-induced dyskinesia in 741 Korean PD patients [19]. They identified a variant in the *GALNT14* gene associated with dyskinesia (odds ratio of 5.5, 95% CI = 2.9–10.3, *p* = 7.88 × 10^−9^), which can potentially predict patients more prone to this complication and may provide glimpses on how to disentangle its pathophysiology. In another study, Prud’hon et al. investigated impulse control disorder (ICD), a significant adverse effect caused by dopamine agonists in PD [20]. Here they compared exome sequencing of two groups of individuals with extreme phenotypes for ICD and found an enrichment of variants in brain-expressed genes of the adenylate cyclase-activating pathway. Using these genes as targets in future studies and clinical trials could lead to better symptomatic treatment options.

There is a growing interest regarding the effect of microbiome on diseases, particularly for PD [21]. Beyond its pathophysiological implications, drug-microbiome interactions can also influence therapeutics. COMT inhibitors, anticholinergics, and levodopa were associated with changes in the microbiome [22]. Gut bacteria, precisely some *Enterococcus* strains carrying the *tdc* gene, can exhibit tyrosine decarboxylase activity, which can convert levodopa to dopamine and decrease the levels of drug in plasma [23]. The amount of the *tdc* gene was correlated with disease duration and higher levodopa doses. Another study found that *Eggerthella* strains can contribute to levodopa degradation, and a single nucleotide variant in this bacteria can predict their enzymatic activity [24]. Interestingly, human decarboxylase inhibitors used in conjunction with levodopa, like carbidopa, do not affect the bacteria enzymatic activity. AFMT, a small-molecule that inhibits bacteria decarboxylase, was suggested as an innovative therapeutic approach. These bacteria or the *tdc* gene may potentially be used as biomarkers to predict or stratify patients who are more responsive to levodopa or more prone to develop levodopa-induced motor complications. This also suggests that the inactivation of the *tdc* gene is a potential future therapeutic target to improve the levodopa response.

Although deep brain stimulation therapy (DBS) is not generally considered a personalized genomics approach, there is evidence that PD patients have varied responses to DBS depending on their genetic background. So far, studies assessing DBS outcomes in patients carrying variants in specific genes are limited in size, but it has been reported that in patients with *LRRK2* variants, outcomes of DBS are similar to cases without known variants [25,26], whereas less favorable outcomes are seen in patients carrying variants in *GBA* [27,28].

As we work towards discovering disease-modifying strategies, it is unlikely that current antiparkinsonian symptomatic treatments, like levodopa and DBS, will lose their importance in the medium term for most patients. However, the goal to achieve a personalized approach for PD is still elusive, in part because evidence from "pre-genomic" era studies is inconclusive. There should be an effort to collect replication cohorts with larger samples and deep phenotyping to derive consistent pharmacogenetics recommendations. The current efforts to increase the power of GWAS for PD risk could also benefit by taking into account the importance of collecting information regarding pharmacological response. Finally, understanding the influence of the microbiome on levodopa metabolism may provide another front to personalize treatments in common forms of PD.

## 3. Genetics Nominates Promising Targets: *LRRK2* and *GBA* Clinical Trials

Despite the remarkable effects of the current treatments and drugs on the symptoms of PD, genetics has played a key role in nominating causative genes or genetic risk factors as targets for different genetic subtypes of PD. Leucine-rich repeat kinase 2 (*LRRK2*) variants are the most common cause of monogenic PD and one of the most common risk factors for idiopathic PD with a variable penetrance between 50–70% [29,30]. The LRRK2 protein exhibits both kinase and GTPase functions, and mounting evidence has shown that known pathogenic *LRRK2* variants increase the kinase activity. The most common PD-linked variant, LRRK2 G2019S, leads to a two-to-threefold increase in kinase activity which is hypothesized to be an underlying molecular mechanism responsible for the development of PD [31]. This gain-of-function implies that utilizing LRRK2 kinase inhibitors may have neuroprotective effects in PD [32,33].

Following positive preclinical experiments, two small molecule inhibitors of LRRK2 developed by Denali Therapeutics, DNL201 and DNL151, are currently in clinical trials [34,35]. A phase 1b, randomized, multicenter, double-blind placebo-controlled clinical trial of DNL201 (NCT04056689) included 29 patients with mild to moderate PD, with or without *LRRK2* variants. The results indicated that levels of LRRK2 phospho Serine-935 and phospho-RAB10 in the blood of PD patients were each decreased by more than 50% at both doses. Meanwhile, a biomarker of lysosomal function, BMP (22:6-bis-monoacylglycero-phosphate), was increased by 20% and 60% in urine at the low and high dose, respectively [36]. Similar trials (NCT04056689) of DNL151 followed and have also met safety and biomarker goals. Given a more flexible dosing regimen, Denali intends to choose DNL151 to advance into phase 2/3 clinical trials in PD patients.

Since genetic studies have indicated no association of LRRK2 loss of function alleles with PD, [37] another approach now entering clinical trials is the use of antisense oligonucleotides (ASOs) to reduce the levels of active LRRK2 protein [38,39]. ASOs are promising therapeutic approaches that aim at directly and chronically decrease LRRK2 kinase activity by editing out the parts of the mRNA known to contain disease associated variants. A phase 1 clinical trial using BIIB094, an ASO to LRRK2, is currently underway to assess its safety, tolerability, and pharmacokinetics in PD patients (NCT03976349). This unique and novel approach is thought to be key to develop a long-term, effective and stable therapeutic treatment decreasing LRRK2 kinase activity and alleviating LRRK2-associated neuronal dysfunction in PD.

As the most common genetic risk factor for PD, *GBA* variants are found in 7–10% of patients with PD [40,41]. Inheriting two copies of defective *GBA* causes Gaucher Disease (GD) with varying severity depending on where the variant is located. Carriers of severe *GBA* variants have an age at onset (AAO) for PD roughly five years earlier and around a three to fourfold increase in PD risk, compared with mild *GBA* variants carriers [42]. Furthermore, severe *GBA* variants appear to be associated with higher risk of cognitive impairment and aggressive cognitive decline [43,44]. There are two common *GBA* variants associated with PD risk which do not cause GD, p.E326K and p.T369M, that may modify GCase activity to a lower level than GD associated variants. It is well established that GD phenotype can also increase the risk for PD [45]. Growing evidence supports the notion that heterozygous PD-related *GBA* variants affect multiple PD pathways [46] (shown in Figure 1) by reducing glucocerebrosidase (GCase) activity in the lysosome, leading to altered lipid metabolism, aggregation of a-synuclein (α-syn) and impaired neuronal transmission. Furthermore, aggregates of α-synuclein inhibit normal GCase activity by restricting GCase transport, thereby causing a pathogenic feedback loop [47]. Current approaches targeting GBA include GCase substrate reduction, gene therapy, small molecule chaperones and enzyme activators.

The MOVES-PD study, a randomized, multicenter, double-blind, placebo-controlled trial, was conducted to evaluate the ability of the glucosylceramide synthase inhibitor Venglustat (GZ/SAR402671) to target substrate reduction in PD patients carrying *GBA* variants (NCT02906020). Part 1 of the phase II trial results revealed that Venglustat safely achieves a dose-dependent reduction of glucosylceramide levels in plasma and cerebrospinal fluid (CSF), however, the most recent earnings report by Sanofi suggests that the trial did not meet the primary goals and has been discontinued. An ongoing Phase 1/2a trial launched by Prevail Therapeutics in early 2020, employs an AAV9-based dosage of PR001A in PD patients with at least one pathogenic *GBA* variant (NCT04127578) to assess its long term (five years) safety and efficacy. A recently reported phase II open label clinical trial of Ambroxol, a GCase chaperone that has previously been used to treat respiratory symptoms, in PD patients with or without GBA variants, demonstrated a decrease in CSF GCase enzyme activity [48]. Although the drug appears safe and well-tolerated, placebo-controlled clinical trials are needed to further confirm their findings. Another single-centre, randomized, double-blind, placebo-controlled trial of Ambroxol is currently in phase II (NCT02914366) [49].

A small molecule activator of GCase (LTI-291) has been under investigation in a phase Ib clinical trial in patients with *GBA* variants conducted by Lysosomal Therapeutics (Trialregister.nl ID: NTR7299). Furthermore, RTB101, an inhibitor of the mammalian target of rapamycin complex 1 (mTORC1), has been tested in a randomized, double-blind, placebo-controlled phase 1b/2a trial of RTB101 alone and in combination with Sirolimus (another inhibitor of mTOR often used an immunosuppressive agent) to be used in PD patients with or without *GBA* variants (ANZCTR ID: ACTR N12619000372189). Interim data from this study revealed that RTB101 was well tolerated and crossed the blood-brain barrier (BBB).

## 4. Genetics as a Tool to Nominate Networks to Be Targeted in Therapeutic Development

Genetics can be used in multiple ways to identify potential genes, proteins, pathways, and networks that may be involved in the pathogenesis of PD and could potentially be therapeutically targeted [50]. The simplest way of identifying targets using genetics is by examining genes known to cause disease or increase risk, like *LRRK2* and *GBA*, using linkage and sequencing studies in families and sporadic cases. Robak et al. expanded this strategy to a larger gene-set using burden analysis in a combination of data from whole exome sequencing (WES) and genotyping of 54 known lysosomal storage disease (LSD) genes to show there is a significant increase in the burden of LSD variants in PD [51]. This association remained significant in multiple cohorts even when GBA was excluded.

Another genetics tool that can be used to select potential therapeutic targets is by identifying variants that are associated with PD risk through GWAS. The latest and largest GWAS meta-analyses have identified over 90 genetic loci harboring common variants that are associated with both PD risk and progression [7,52,53]. Burden analyses examining coding variants are now regularly combined with GWAS results to prioritize genes at a locus that is associated with PD [54]. However, the non-coding portion of the genome is significantly larger than protein coding regions so it is unusual that a specific gene is identified by GWAS. This makes nomination of specific therapeutic targets at a GWAS locus very difficult [55]. In general, the effects exerted by individual GWAS variants are quite small, but when they are combined to determine a polygenic risk score (PRS) they can be used to further stratify cases from low to high risk [56,57,58]. PRS is defined as a model that sums the contribution of multiple risk variants of variable magnitude of effect, as determined by GWAS summary statistics. The 90 risk loci identified in the most recent PD meta-analysis are associated with higher relative risk of developing PD, with those individuals in the top 10% of PRS being nearly six-fold more likely to develop PD than those in the bottom 10% [7]. In the first major study on PRS in PD, Ibanez et al. showed that PRS in cases, excluding variants in known familial or risk genes, associated with PD status and age at onset but not with the levels of three predicted CSF biomarkers [56].

Instead of focusing on a single variant or PRS, genetic data can be integrated with transcriptomic, proteomic and protein-protein interaction (PPI) networks to nominate affected biological pathways that a single data type might miss [59,60]. A recent study examined the association of 2199 pre-defined gene sets grouped by biological process with PD by assigning a Polygenic Effect Score (PES) to each gene-set and then performing an association study [59]. The authors identified a wide range of gene-sets that were associated with PD. Further analysis using Mendelian randomization in genome-wide expression and methylation datasets identified genes with quantitative trait loci (QTL) for expression in blood and brain, as well as changes in methylation at multiple CpG sites that are associated with PD risk. This unbiased and data-driven study provided a foundational resource for the PD community through a publicly available pathways browser. Pathways previously implicated by genetics and functional studies also found to be significant in this study include endocytic trafficking [61,62], autophagic-lysosomal function [51,63], mitochondrial function [64,65], protein aggregation [66], neuronal transmission [67], lipid metabolism [68,69], and certain inflammatory pathways [70,71] (Figure 1). It has also been shown that similar pathways can be deficient in both familial and common forms of PD [51,62] and multiple networks can overlap or a single pathway can act alone. Interestingly, some of the nominated gene-sets span the etiological risk spectrum in which both common and rare variation contribute to PD susceptibility.

Combining all genomic, transcriptomic and proteomic data to identify affected pathways in PD allows individuals without variants in known risk factors to be stratified by the pathways thought to be involved in their subtype of disease. Examining pathways instead of genes also suggests that other members of the pathway could be used as therapeutic targets even if the associated gene is not druggable (Figure 1). For example, the RTB101/rapamycin clinical trial described previously (ANZCTR ID: ACTR N12619000372189) targets the mTOR complex, which is not itself associated with PD, but is involved in a signaling pathway that regulates autophagy and has been shown to rescue dopaminergic neuron degeneration in some PD models [72]. Examining the genetic networks associated with PD and employing drug repositioning to target them may be a way to quickly increase the number of PD therapeutics available in the future.

The integration of genetic (like GWAS) and transcriptomic (e.g., RNA-sequencing) data can further inform the development of personalized medicine for the diagnosis and treatment of PD. These two data types can yield biological insight into candidate genes and pathways for the development of targeted therapeutics. When multi-omics data types such as these are combined, we can begin to gain mechanistic insights. Recent studies have aimed at linking the genes underlying GWAS loci to functional consequences by leveraging large-scale transcriptomic datasets to prioritize genes by using a transcriptome-wide association study (TWAS) [73]. Another way to integrate these data types uses colocalization and weighted gene coexpression network analysis to identify candidate genes [74]. These comprehensive and unbiased explorations provide a strong foundation for further mechanistic studies that can help functionally characterize therapeutic targets and plan clinical trials.

## 5. Genetics Informs Parkinson’s Disease Subtyping

Understanding the etiological heterogeneity of PD is widely recognized as a critical step in achieving personalized and disease-modifying approaches. The first attempts to subtype PD used clinical information, like age at onset. In fact, early-onset patients, as compared to late-onset, tend to exhibit a slower disease progression, less severe clinical course and a higher risk of developing levodopa-induced dyskinesia [75]. Subtyping PD according to motor and non-motor symptoms is also a common approach, either using pre-defined clinical criteria or a data-driven approach. Its utility has been questioned since the first strategy does not seem to be stable along the disease course, and the latter lacks reproducibility [76,77]. Despite these limitations, a subgroup of PD with tremor-dominant symptoms is widely recognized, in opposition to a group with less tremor and more akinetic and gait dysfunction [78]. The next frontier to delineate PD heterogeneity must incorporate more objective measures such as biomarkers and deep-phenotyping information to define biological subtypes suitable for personalized interventions [79].

Developing strategies for diagnosis of the prodromal phase of PD and identifying biomarkers that are able to measure its progression are essential in the search for new therapies. Studies suggest that by the time of diagnosis, patients already show a neuronal loss of 40–50% in the substantia nigra [80,81], explaining, at least in part, why previous trials have failed to find a disease-modifying effect [82,83]. Since 2015, the Movement Disorder Society has been proposing diagnostic research criteria to define prodromal PD [84]. Multiple clinical symptoms were included, like REM sleep behavior disorder, olfactory loss, constipation, and depression. In 2019, the criteria were updated [85], and genetics is now combined with clinical and other types of biomarkers to improve PD prediction. Carriers of rare highly-penetrant and pathogenic variants (like those in *SNCA*, *PRKN*, and *PINK1*) formed distinct prodromal monogenic PD subgroups. Variants of intermediate magnitude of effect in genes such as *GBA* and *LRRK2* were included considering their age-dependent penetrance. Finally, for common variants with low individual effect identified in previous GWAS studies, the criteria recommend calculating the PRS for a large sample series with genetic data and classifying patients according to the risk score distribution in the sample. A recent study has identified common non-coding SNPs within *GBA* regulating *GBA* expression in peripheral tissues [86]. Interestingly, the authors report that non-coding SNPs within *GBA* also coregulate potential modifier genes in the central nervous system and/or peripheral tissues, delaying disease onset by 5 years. Although the nominated variants need to be functionally validated, this promising approach opens the door for future disease stratification, personalized drug selection and the possible development or repurposing of novel drugs.

## 6. Conclusions and Future Directions

Symptomatic treatment with levodopa has been the norm in PD for more than fifty years despite its sometimes serious side effects. Until recently, efforts to improve treatment options have been slow. Although there is still much to learn about the molecular mechanisms underlying PD, significant progress is now being made towards the identification of potential therapeutic targets in this complex disease. Genetics has played a key role in increasing the number of recent and ongoing clinical trials. The random genetic assortment of patients in clinical trials represents an avoidable source of variance that is likely contributing to the high failure rate seen in PD trials. In fact, even within specific subgroups carrying known PD variants, large variation between patients still exists. Different variants within a specific gene can lead to differential effects on PD phenotypes, and as previously shown [87], this genetic imbalance affects clinical trial design. Acknowledging the limitation that understanding the exact effect of all human genetic variation on disease aetiology and drug response is not yet possible, at the minimum, balancing known disease risk variants should be performed. Using PRS to stratify patients by low and high risk could help identify drugs that will work in some forms of PD.

Stratified trial designs can be used to potentially increase the efficiency of a trial. This was evident in exemplary form in the relevant success attributable to the enrollment strategy of the Aducanumab trial in 2015 and deviations from this strategy being potentially related to less positive results in more recent development phases of the drug [88]. Using genetic, clinical, imaging or other molecular biomarkers to enroll patients that may have a higher probability to efficiently respond to an intervention is key to trial success and a central concept in stratified trials. Another aspect of using potential patient stratification to design more efficient trials, particularly in degenerative type diseases, is to identify patients early in the disease course where targetable cell types of interest are still functional or available and may be protected or rescued; too late in disease course irreparable or immutable damage may have already occurred.

Additionally, advancing target development by combining genomic, transcriptomic and proteomic data has broadened the search space for potential drugs. Focusing not just on monogenic or known risk factors but also the various pathways and networks implicated across the subtypes of idiopathic PD may soon increase the available therapeutic options. The numerous studies directed by genetics described here show that the age of personalized medicine in PD is fast approaching.

## Figures and Tables

**Figure 1 jpm-11-00169-f001:**
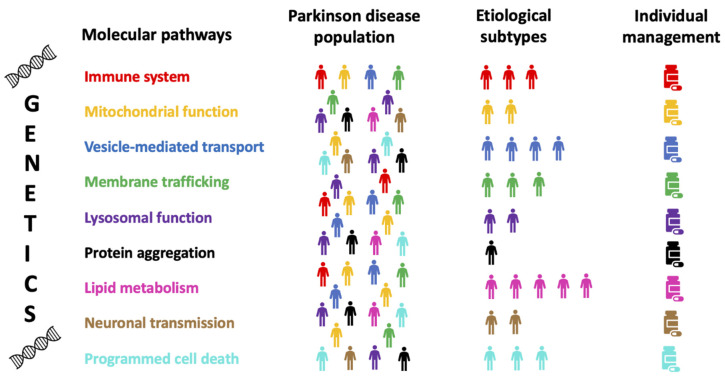
The role of genetics to define subtypes of Parkinson’s disease and to develop potential etiological based interventions. This figure serves as an example showing that certain molecular pathways within the same disease can be significantly enriched in different individuals. These pathways may eventually become targets for personalized based interventions.

## Data Availability

Not applicable.

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
