# Peer review of "Advancing Personalized Medicine in Common Forms of Parkinson’s Disease through Genetics: Current Therapeutics and the Future of Individualized Management"

_jpm, 2021, doi:10.3390/jpm11030169_

Round 1
Reviewer 1 Report
Quite a number of reviews related to therapeutic innovations in Parkinson's have recently appeared, however this manuscript takes an interesting perspective, choosing to focus more broadly on a overview of future genetically stratified and targeted therapies, both symptomatic and neuroprotective.
That said, the manuscript falls down in a number of keys areas and I do not think can be published without heavy editing and the addition of a significant amount of content. Broadly the following areas need significant attention
1) Although the manuscript starts by talking about the genetic heterogenicity of PD, it then fails to talk about most of it. There is no mention of snca, pink, prkn DJ1 or other important monogenic forms of PD. heterozygosity of PRKN for instance may be a genetic risk factor for PD in itself. Moreover it does not give mentions of the broad genetic landscape of PD or the missing heritability which is at the heart of most current PD genetic research.
2) GBA and LRRK2 are talked of as 'monogenic forms' of PD. Both are genetic risk factors for PD. LRRK2 has a penetrance of 50 to 70%. GBA has a penetrance of around 10%. In the case of GBA this means the vast majority of GBA carriers will not develop PD which is vital if you are talking about startifying therapies
3) disease causing variants of GBA (and to a lesser extent LRRK2) are highly heterogenous. There is no discussion of 'severe' and mild GBA mutations or the non gaucher causing PD risk variants (E326K and T369M). This is key, because the odds rations of developing PD vary from 2 to 20 depending on the variant in question.
5) There is no discussion of the histopathology and cell biology of PD. The synuclein centric model and rationale for the assumption that alpha synuclein aggregation is the disease causing mechanism is key to understanding the utility of many therapies. Equally the fact that some genetic forms of PD do not cause lewy bodies emphasises that multiple overlapping disease aetiologies are at play.
4) The authors spend some time talking about dopaminergic therapy, however they miss the key point, which is that initially dopaminergic therapy is on the whole highly effective, but as the disease progresses and as dopaminergic neurons die, higher doses are require. This eventually leads to a lack of response to levo dopa and side effects due to the ever higher doses being used, most prominently dyskinsia. Equally Pd affects many neuronal subtypes, including cholinergic neurons, which may explain the why a portion of patients do not respond to dopaminergic therapy.
5) There is no discussion of the prodromal phase of PD, which is key, as screening and stratifying before motor symptom onset will be key to avoiding disability for any neuroprotective (see MDS prodromal PD diagnostic criteria).
6) PD is clinically very heterogenous and many attempts have been made to define subtypes of it. Whilst these attempts have on the whole been lacklustre, they do need discussion. Equally genetic forms of PD do have different characteristics (GBA has a more cogntiive phenotype and faster progression, LRRK2 often has more tremor, PRKN seems to have more dystonia).
7) There needs to be more discussion of other (advanced) PD therapies, such as DBS. Understanding who will benefit from these therapies is important, for instance a number of papers have shown GBA carriers have a worse outcome from DBS.
8) A number therapeutic approaches, such a ASOs and gene therapy are discussed but with little detail. The biological rationale of these therapies needs to be outlined
9) genetic variability is a major confounding issue in clinical trial design in PD and needs discussion
10) The section on polygenic risk needs expansion and more explanation
11) The paper talks a fair built about GWAS with little explanation as to what it is. GWAS identifies loci, it cannot look at individual genes because it uses linkage disequilibrium in impute large portions of the genome, based on a relatively small numbers of SNPs covering the genome. GWAS has confirmed the role of candidate pathways and identified other loci of interest which then needs to be defined using other techniques. It is one of a number of useful tools, not a means unto itself. It is very good at looking for high frequency low penetrance effects, but much poorer at looking for for less common variants. GBA and LRRK2 were both identified by clinical aptitude.
12) The written style is at times waffly and repetitive, particularly in the introduction). It needs more focus. At times it would benefit from the attention of a native english speaker.
13) The article talks about how stratification influenced trial design but then fails describe this in any detail.
14) In general there needs to be more detail associated with each reference and an outline of the specifics of why they are referenced. A number of reviews are also referenced which is sloppy in a review article.
There are also a number of specific issues:
1) The article refers to mutations rather than variants, which is the more current terminology
2) reference 31: GCase activity decreased in CSF, CSF GCase protein levels increased!
3)I'm not quite clear what figure 1 adds to the article as some of the themes it displays are barely mentioned in the text
Author Response
Quite a number of reviews related to therapeutic innovations in Parkinson's have recently appeared, however this manuscript takes an interesting perspective, choosing to focus more broadly on a overview of future genetically stratified and targeted therapies, both symptomatic and neuroprotective.
That said, the manuscript falls down in a number of keys areas and I do not think can be published without heavy editing and the addition of a significant amount of content. Broadly the following areas need significant attention
1) Although the manuscript starts by talking about the genetic heterogenicity of PD, it then fails to talk about most of it. There is no mention of snca, pink, prkn DJ1 or other important monogenic forms of PD. heterozygosity of PRKN for instance may be a genetic risk factor for PD in itself. Moreover it does not give mentions of the broad genetic landscape of PD or the missing heritability which is at the heart of most current PD genetic research.
We thank the reviewer 1 for their comments.
Since monogenic forms of Parkinson’s disease have been widely covered in some of our previous reviews (see Reed et al., 2018. The role of monogenic genes in idiopathic Parkinson's disease; Bandres-Ciga et al., 2019. Genetics of Parkinson's disease: An introspection of its journey towards precision medicine), we focused the current review on common genetics forms of the disease. We have stated it as follows:
“On the whole, pure monogenic forms of PD are rare and although variants in genes like SNCA, PINK1, PRKN and DJ1 are well established causes of disease it would be difficult to collect enough patients to create an appropriately powered clinical trial in these populations. For this reason in this review we will focus on more common forms of disease including those with variants in known risk factors, like LRRK2 and GBA, as well as idiopathic forms of PD where the exact cause is not known but it is thought to be a combination of genetic and environmental factors. Current estimates of PD heritability have revealed that the contribution of genetic factors to PD phenotype is about 22% indicating that stratifying patients by genomic factors is possible [7][8].”
2) GBA and LRRK2 are talked of as 'monogenic forms' of PD. Both are genetic risk factors for PD. LRRK2 has a penetrance of 50 to 70%. GBA has a penetrance of around 10%. In the case of GBA this means the vast majority of GBA carriers will not develop PD which is vital if you are talking about startifying therapies
We thank reviewer 1 for this comment that we have amended as follows:
“Despite the remarkable effects of the current treatments and drugs on the symptoms of PD, genetics has played a key role in nominating causative genes or genetic risk factors as targets for different genetic subtypes of PD. Leucine-rich repeat kinase 2 (LRRK2) variants are the most common cause of monogenic PD and one of the most common risk factors for idiopathic PD with a variable penetrance between 50-70% [29][30].”
“As the most common genetic risk factor for PD, GBA variants are found in 7-10% of patients with PD [39][40]. Inheriting two copies of defective GBA causes Gaucher Disease (GD) with varying severity depending on where the variant is located…”
3) disease causing variants of GBA (and to a lesser extent LRRK2) are highly heterogenous. There is no discussion of 'severe' and mild GBA mutations or the non gaucher causing PD risk variants (E326K and T369M). This is key, because the odds rations of developing PD vary from 2 to 20 depending on the variant in question.
We thank reviewer 1 for this comment and we have amended as follows:
“Carriers of severe GBA variants have an age at onset (AAO) for PD roughly 5 years earlier and around a three to fourfold increase in PD risk, compared with mild GBA variants carriers [41]. Furthermore, severe GBA variants appear to be associated with higher risk of cognitive impairment and aggressive cognitive decline [42][43]. There are two common GBA variants associated with PD risk which do not cause GD, p.E326K and p.T369M, that may modify GCase activity to a lower level than GD associated variants. It is well established that GD phenotype can also increase the risk for PD [44]. Growing evidence supports the notion that heterozygous PD-related GBA variants affect multiple PD pathways [45] (shown in Figure 1) by reducing glucocerebrosidase (GCase) activity in the lysosome, leading to altered lipid metabolism, aggregation of a-synuclein (α-syn) and impaired neuronal transmission.”
5) There is no discussion of the histopathology and cell biology of PD. The synuclein centric model and rationale for the assumption that alpha synuclein aggregation is the disease causing mechanism is key to understanding the utility of many therapies. Equally the fact that some genetic forms of PD do not cause lewy bodies emphasises that multiple overlapping disease aetiologies are at play.
We thank reviewer 1 for this insightful comment which we have addressed as follows:
“The link of PD to α-synuclein was the first decisive proof of a genetic defect leading to disease [2]. Later on, the first PD GWAS identified SNCA [3] as one of the major genes driving risk for sporadic PD, linking both familial and sporadic forms. Abnormal α-synuclein is a histopathological hallmark of PD patients, but also patients with other neurodegenerative conditions, collectively termed synucleinopathies, making this target promising. However, the fact that PD patients harbouring genetic defects in genes such as PRKN do not present with Lewy body pathology, strengthens the notion that distinct entities and multiple overlapping etiologies are at play.”
4) The authors spend some time talking about dopaminergic therapy, however they miss the key point, which is that initially dopaminergic therapy is on the whole highly effective, but as the disease progresses and as dopaminergic neurons die, higher doses are require. This eventually leads to a lack of response to levo dopa and side effects due to the ever higher doses being used, most prominently dyskinsia. Equally Pd affects many neuronal subtypes, including cholinergic neurons, which may explain the why a portion of patients do not respond to dopaminergic therapy.
We thank reviewer 1 for this comment. We improved the text accordingly:
"The symptomatic treatment available for PD targets the motor symptoms induced by the dopaminergic deficit due to the degeneration of the substantia nigra. Nevertheless, the disease affects other systems and regions in the brain, which leads to a myriad of levodopa-resistant motor and non-motor symptoms for which we do not have well-established pharmacological interventions. Despite this limitation, PD is the only neurodegenerative disorder with a symptomatic treatment that provides a substantial benefit. Since the introduction of levodopa in the 1960s, it has changed the natural history of PD and remains the gold standard of treatment [9]. However, the pharmacological response is variable and, as the disease progresses, higher doses of levodopa are required. Also, complications induced by chronic treatment can develop over time, including motor fluctuations and dyskinesia, which affect almost half of the patients after five years of treatment and nearly all in the long term [10][11][12]. This situation impairs the patient’s quality of life and demands more costly and complex therapeutic regimens.“
5) There is no discussion of the prodromal phase of PD, which is key, as screening and stratifying before motor symptom onset will be key to avoiding disability for any neuroprotective (see MDS prodromal PD diagnostic criteria).
We thank the reviewer for this comment which we have addressed as follows:
“Understanding the etiological heterogeneity of PD is widely recognized as a critical step in achieving personalized and disease-modifying approaches. The first attempts to subtype PD used clinical information, like age at onset. In fact, early-onset patients, as compared to late-onset, tend to exhibit a slower disease progression, less severe clinical course and a higher risk of developing levodopa-induced dyskinesia [74]. Subtyping PD according to motor and non-motor symptoms is also a common approach, either using pre-defined clinical criteria or a data-driven approach. Its utility has been questioned since the first strategy does not seem to be stable along the disease course, and the latter lacks reproducibility [75][76]. Despite these limitations, a subgroup of PD with tremor-dominant symptoms is widely recognized, in opposition to a group with less tremor and more akinetic and gait dysfunction [77]. The next frontier to delineate PD heterogeneity must incorporate more objective measures such as biomarkers and deep-phenotyping information to define biological subtypes suitable for personalized interventions [78].
Developing strategies for diagnosis of the prodromal phase of PD and identifying biomarkers that are able to measure its progression are essential in the search for new therapies. Studies suggest that by the time of diagnosis, patients already show a neuronal loss of 40-50% in the substantia nigra [79][80], explaining, at least in part, why previous trials have failed to find a disease-modifying effect [81][82]. Since 2015, the Movement Disorder Society has been proposing diagnostic research criteria to define prodromal PD [83]. Multiple clinical symptoms were included, like REM sleep behavior disorder, olfactory loss, constipation, and depression. In 2019, the criteria was updated [84], and genetics is now combined with clinical and other types of biomarkers to improve PD prediction. Carriers of rare highly-penetrant and pathogenic variants (like those in SNCA, PRKN, and PINK1) formed distinct prodromal monogenic PD subgroups. Variants of intermediate magnitude of effect in genes such as GBA and LRRK2 were included considering their age-dependent penetrance. Finally, for common variants with low individual effect identified in previous GWAS studies, the criteria recommends calculating the PRS for a large sample series with genetic data and classifying patients according to the risk score distribution in the sample.”
6) PD is clinically very heterogenous and many attempts have been made to define subtypes of it. Whilst these attempts have on the whole been lacklustre, they do need discussion. Equally genetic forms of PD do have different characteristics (GBA has a more cogntiive phenotype and faster progression, LRRK2 often has more tremor, PRKN seems to have more dystonia).
We thank reviewer 1 for this comment. We have already addressed these issues in questions 3, 5 and 7.
7) There needs to be more discussion of other (advanced) PD therapies, such as DBS. Understanding who will benefit from these therapies is important, for instance a number of papers have shown GBA carriers have a worse outcome from DBS.
We thank reviewer 1 for this comment that we have addressed as follows:
“Although deep brain stimulation therapy (DBS) is not generally considered a personalized genomics approach, there is evidence that PD patients have varied responses to DBS depending on their genetic background. So far, studies assessing DBS outcomes in patients carrying variants in specific genes are limited in size, but it has been reported that in patients with LRRK2 variants, outcomes of DBS are similar to cases without known variants [25][26], whereas less favorable outcomes are seen in patients carrying variants in GBA [27][28].”
8) A number therapeutic approaches, such a ASOs and gene therapy are discussed but with little detail. The biological rationale of these therapies needs to be outlined
We have further expanded this section as follows:
“ Since genetic studies have indicated no association of LRRK2 loss of function alleles with PD,
[36] another approach now entering clinical trials is the use of antisense oligonucleotides (ASOs) to reduce the levels of active LRRK2 protein [37][38]. ASOs are promising therapeutic approaches that aim at directly and chronically decrease LRRK2 kinase activity by editing out the parts of the mRNA known to contain disease associated variants. A phase 1 clinical trial using BIIB094, an ASO to LRRK2, is currently underway to assess its safety, tolerability, and pharmacokinetics in PD patients (NCT03976349). This unique and novel approach is thought to be key to develop a long-term, effective and stable therapeutic treatment decreasing LRRK2 kinase activity and alleviating LRRK2-associated neuronal dysfunction in PD.”
9) genetic variability is a major confounding issue in clinical trial design in PD and needs discussion
We thank reviewer 1 for this comment. We have widely discussed it as follows:
“The random genetic assortment of patients in clinical trials represents an avoidable source of variance that is likely contributing to the high failure rate seen in PD trials. In fact, even within specific subgroups carrying known PD variants, large variation between patients still exists. Different variants within a specific gene can lead to differential effects on PD phenotypes, and as previously shown [85], this genetic imbalance affects clinical trial design. Acknowledging the limitation that understanding the exact effect of all human genetic variation on disease aetiology and drug response is not yet possible, at the minimum, balancing known disease risk variants should be performed. Using PRS to stratify patients by low and high risk could help identify drugs that will work in some forms of PD.”
10) The section on polygenic risk needs expansion and more explanation
We have expanded this section as follows:
“PRS is defined as a model that sums the contribution of multiple risk variants of variable magnitude of effect, as determined by GWAS summary statistics. The 90 risk loci identified in the most recent PD meta-analysis are associated with higher relative risk of developing PD, with those individuals in the top 10% of PRS being nearly six-fold more likely to develop PD than those in the bottom 10% [7]. In the first major study on PRS in PD, Ibanez et al. showed that PRS in cases, excluding variants in known familial or risk genes, associated with PD status and age at onset but not with the levels of three predicted CSF biomarkers [55].”
11) The paper talks a fair built about GWAS with little explanation as to what it is. GWAS identifies loci, it cannot look at individual genes because it uses linkage disequilibrium in impute large portions of the genome, based on a relatively small numbers of SNPs covering the genome. GWAS has confirmed the role of candidate pathways and identified other loci of interest which then needs to be defined using other techniques. It is one of a number of useful tools, not a means unto itself. It is very good at looking for high frequency low penetrance effects, but much poorer at looking for for less common variants. GBA and LRRK2 were both identified by clinical aptitude.
We thank the reviewer for this comment. However, we have widely covered the concept and application of GWAS in previous reviews (Bandres-Ciga et al., 2019. Genetics of Parkinson's disease: An introspection of its journey towards precision medicine). The scope of this review was to provide an overview of current therapeutics and the future of individualized management in Parkinson’s disease. However, in an attempt to address the reviewer’s concern, we include the following information:
“The latest and largest GWAS meta-analyses have identified over 90 genetic loci harboring common variants that are associated with both PD risk and progression [7][51][52]. Burden analyses examining coding variants are now regularly combined with GWAS results to prioritize genes at a locus that is associated with PD [53]. However, the non-coding portion of the genome is significantly larger than protein coding regions so it is unusual that a specific gene is identified by GWAS. This makes nomination of specific therapeutic targets at a GWAS locus very difficult [54]. In general, the effects exerted by individual GWAS variants are quite small, but when they are combined to determine a polygenic risk score (PRS) they can be used to further stratify cases from low to high risk [55][56][57].”
12) The written style is at times waffly and repetitive, particularly in the introduction). It needs more focus. At times it would benefit from the attention of a native english speaker.
We thank the reviewer and apologize for the lack of clarity. We have carefully edited it and the content has been reviewed by a native english speaker.
13) The article talks about how stratification influenced trial design but then fails describe this in any detail.
We have further expanded this section as follows:
“Stratified trial designs can be used to potentially increase the efficiency of a trial. This was evident in exemplary form in the relevant success attributable to the enrollment strategy of the Aducanumab trial in 2015 and deviations from this strategy being potentially related to less positive results in more recent development phases of the drug [https://www.alzforum.org/therapeutics/aducanumab]. Using genetic, clinical, imaging or other molecular biomarkers to enroll patients that may have a higher probability to efficiently respond to an intervention is key to trial success and a central concept in stratified trials. Another aspect of using potential patient stratification to design more efficient trials, particularly in degenerative type diseases, is to identify patients early in the disease course where targetable cell types of interest are still functional or available and may be protected or rescued; too late in disease course irreparable or immutable damage may have already occured.”
14) In general there needs to be more detail associated with each reference and an outline of the specifics of why they are referenced. A number of reviews are also referenced which is sloppy in a review article.
We thank the reviewer and have carefully addressed this point in the text.
There are also a number of specific issues:
1) The article refers to mutations rather than variants, which is the more current terminology
We apologize for this oversight and have amended it accordingly.
2) reference 31: GCase activity decreased in CSF, CSF GCase protein levels increased!
We apologize for this mistake and have amended it.
3)I'm not quite clear what figure 1 adds to the article as some of the themes it displays are barely mentioned in the text
We have included a legend to the figure and expanded the main text:
Legend:
“This figure serves as an example showing that certain molecular pathways within the same disease can be significantly enriched in different individuals. These pathways may eventually become targets for personalised based interventions.”
Manuscript:
“Instead of focusing on a single variant or PRS, genetic data can be integrated with transcriptomic, proteomic and protein-protein interaction (PPI) networks to nominate affected biological pathways that a single data type might miss [58][59]. A recent study examined the association of 2,199 pre-defined gene sets grouped by biological process with PD by assigning a Polygenic Effect Score (PES) to each gene-set and then performing an association study [58]. The authors identified a wide range of gene-sets that were associated with PD. Further analysis using Mendelian randomization in genome-wide expression and methylation datasets identified genes with quantitative trait loci (QTL) for expression in blood and brain as well as changes in methylation at multiple CpG sites that are associated with PD risk. This unbiased and data-driven study provided a foundational resource for the PD community through a publicly available pathways browser. Pathways previously implicated by genetics and functional studies also found to be significant in this study include endocytic trafficking [60] [61], autophagic-lysosomal function [50][62], mitochondrial function [63] [64], protein aggregation [65], neuronal transmission [66], lipid metabolism [67] [68], and certain inflammatory pathways [69][70] (Figure 1). It has also been shown that similar pathways can be deficient in both familial and common forms of PD [61][50] and multiple networks can overlap or a single pathway can act alone. Interestingly, some of the nominated gene-sets span the etiological risk spectrum in which both common and rare variation contribute to PD susceptibility.
Combining all genomic, transcriptomic and proteomic data to identify affected pathways in PD allows individuals without variants in known risk factors to be stratified by the pathways thought to be involved in their subtype of disease. Examining pathways instead of genes also suggests that other members of the pathway could be used as therapeutic targets even if the associated gene is not druggable (Figure 1).”

Reviewer 2 Report
Reed and colleagues provide a summary of the potential use of personalized medicine for the diagnosis and treatment of PD in light of the current development of PD's genetic studies. The review is clear, albeit a bit short. My main recommendation to the authors is to expand this review by providing more discussion in the following areas:
- Provide an additional discussion of the advances made in the understanding of transcriptome changes that are associated with PD.
- A corollary will be important to discuss how the integration of genetic (GWAS) and transcriptomic (e.g., RNA-sequencing) data can further inform the development of personalized medicine for the diagnosis and treatment of PD.
- It is important to discuss how the personalized approaches for PD treatment may differ between patients with early-onset PD.
Round 2
Reviewer 1 Report
The manuscript is much improved and nearly ready for publication. There are a number of new points that should be addressed following these revisions.
1) The MOVES-PD trial has just released results which show it failed to meet its primary outcome. please amend.
2) PRO.MED.CSA provided the IMP free of trial for the ambroxol trial, however this was an academic clinical trial lead by an independent research group. The text implies it was a commercial trial, which it was not.
3) The following paper is an important attempt to define genetic subtypes of PD based on haplotype and should be commented on:
https://movementdisorders.onlinelibrary.wiley.com/doi/full/10.1002/mds.28144
Schierding et al. 2020 Movement DisordersAuthor Response
Please see the attachment.

Reviewer 2 Report
I do not have any more comments for the authors.
Author Response
We thank Reviewer 2 for the positive feedback